# Early Detection of the Pathogenetic Variants of Homologous Recombination Repair Genes in Prostate Cancer: Critical Analysis and Experimental Design

**DOI:** 10.3390/biology14020117

**Published:** 2025-01-23

**Authors:** Irene Bottillo, Alessandro Sciarra, Giulio Bevilacqua, Alessandro Gentilucci, Beatrice Sciarra, Valerio Santarelli, Stefano Salciccia, Francesca Bacigalupo, Francesco Pastacaldi, Maria Pia Ciccone, Laura De Marchis, Daniele Santini, Fabio Massimo Magliocca, Elisabetta Merenda, Flavio Forte, Paola Grammatico

**Affiliations:** 1Division of Medical Genetics, Department of Experimental Medicine, San Camillo-Forlanini Hospital, Sapienza University, 00161 Rome, Italy; irene.bottillo@uniroma1.it (I.B.); francesca.bacigalupo99@gmail.com (F.B.); fpastacaldi@yahoo.it (F.P.); mariapia.ciccone@uniroma1.it (M.P.C.); paola.grammatico@uniroma1.it (P.G.); 2Department “Materno Infantile e Scienze Urologiche”, University Sapienza, Viale Policlinico 155, 00161 Rome, Italy; giulio.bevilacqua@uniroma1.it (G.B.); alegenti@yahoo.it (A.G.); va.santarelli@uniroma1.it (V.S.); stefano.salciccia@uniroma1.it (S.S.); 3Department of Pharmaceutic Chemistry, University Sapienza, 00161 Rome, Italy; beatrice.sciarra@uniroma1.it; 4Department Oncology, University Sapienza, 00161 Rome, Italy; laura.demarchis@uniroma1.it (L.D.M.); daniele.santini@uniroma1.it (D.S.); 5Department of Pathology, University Sapienza, 00161 Rome, Italy; fabiomassimo.magliocca@uniroma1.it (F.M.M.); elisabetta.merenda@uniroma1.it (E.M.); 6Urology Unit, Vannini Hospital, 00190 Rome, Italy; flavioforte@hotmail.com

**Keywords:** prostate neoplasm, BRCA gene, DDR gene

## Abstract

To date, in prostate cancer (PCa) patients, the analysis of pathogenic variants (PVs) in DNA Damage Response (DDR) genes remains complex, time-consuming, and at high cost, so that it is often limited to a single gene. The predictive value of a more extensive analysis of a large pool of PVs in the DDR family could offer prognostic indications, which is particularly useful in the earlier phase of PCa disease. The high complexity and costs of this type of analysis can limit its use in clinical practice, and a selection of high-risk patients or those based on family histories is certainly necessary. In our experience, we detected oncogenetic and likely oncogenetic variants in the *ATM*, *BRCA1*, *PTEN*, *KMT2D*, and *CDH1* genes, as well as in non-metastatic PCa cases.

## 1. Introduction

The treatment of prostate cancer (PCa) has undergone an epochal positive evolution in recent years. In particular, this evolution has led to the main concept of anticipation and intensification of therapy in different phases of PCa progression. The point of failure of this process, which is accentuated due to the numerous therapeutic options that these patients have at their disposal, is the identification of valid prognostic indicators of therapy response and the construction of a tailored precision therapy for each specific patient [1,2,3].

The etiology of prostate cancer (PCa) is driven by a multifaceted interplay of genetic and environmental factors. At the molecular level, one key aspect of this process involves genes responsible for DNA Damage Response (DDR), which play a central role in maintaining genomic integrity [1]. These DDR genes facilitate the detection and repair of DNA damage, such as single-strand breaks (SSBs) and double-strand breaks (DSBs), which can accumulate during cell division. Their function is critical in preventing the accumulation of mutations that could lead to tumorigenesis, as well as inducing apoptosis in cells harboring potentially harmful mutations [2,3,4,5].

Mutations in DDR genes, particularly those in pathogenic variants (PVs), compromise the efficiency of DNA repair mechanisms, thereby heightening the risk of genomic instability. When DDR systems are impaired, the cell becomes more susceptible to the accumulation of DNA damage, which can result in the uncontrolled growth of cancerous cells. In the context of DDR, the poly-ADP-ribose polymerase (PARP) system is primarily involved in detecting and repairing SSBs, while the homologous recombination repair (HRR) pathway plays a dominant role in the repair of DSBs [4,5,6]. The pathogenic mutations in HRR genes determine the HRR deficiency (HRD) status, mainly caused by single-nucleotide variants and insertions/deletions. In particular, the PVs of HRR genes can be visible as genomic scars, including loss of heterozygosity (LOH), telomeric allelic imbalance (TAI), and large-scale state transitions (LSTs) [4,5].

*BRCA1* and *BRCA2*, the most well-known genes within the HRR pathway, are essential for maintaining genomic stability, and their PVs are strongly associated with an elevated risk of a spectrum of cancers, including breast, ovarian, prostate, pancreatic, and colon cancers. Such insights may offer opportunities for more precise therapeutic strategies, including the potential for targeted therapies that exploit DDR deficiencies in cancer cells [1,7].

BRCA mutations in several neoplasms have been associated with short metastatic-free survival, short cancer-specific survival, and the prediction of response to PARP inhibitors [7].

HRR mutations in tumor cells induce a dependency for single-strand break reparation through the PARP system, providing the rationale to develop PARP inhibitors. PARP inhibitors block the PARP system, causing an accumulation of DNA damage in DDR-defective tumor cells [6,7].

The importance of the *BRCA1/2* test in patients with PCa is justified by several reasons: it has been shown that the PVs of the BRCA genes, whether of a germinal or somatic nature, represent a predictive biomarker of greater sensitivity to treatment with inhibitors of the enzyme PARP in patients with hormone-resistant metastatic prostate cancer [1,6,7]. For patients with PCa who are positive for BRCA germline PVs, appropriate surveillance programs should be indicated to manage the risk of developing second cancers associated with BRCA PVs [2,3,7].

In general, the detection of PVs of the BRCA genes in PCa patients can help to define the patient’s prognosis and the choice of the therapeutic procedure [2,3].

To date, the analysis of PVs of HRR genes is recommended in patients with metastatic prostate cancer and is linked to precision therapy with PARP inhibitors. Therapy with PARP inhibitors is recommended for metastatic castration-resistant PCa (mCRPC) patients as second-line monotherapy or first-line monotherapy in association with androgen receptor pathway inhibitors (ARPIs) [2,3,6,8].

## 2. DDR Analysis: Prognostic Role in Different Stages of PCa

### 2.1. Incidence

The incidence of PVs in various DDR genes among PCa patients has been reported in multiple studies and registries. Among men with metastatic PCa (mPCa), the frequency of somatic PVs in DDR genes ranges from 21% to 30%, a significantly higher rate compared to non-metastatic PCa (nmPCa), where the incidence is between 5% and 10% [9,10,11,12]. In non-metastatic PCa, *BRCA2* mutations are the most common, occurring in 2.5–5.0% of cases, followed by mutations in *ATM* (1.0–3.5%) and *BRCA1* (0.5–1.0%). In metastatic PCa, *BRCA2* mutations account for 11–15%, *ATM* mutations for 5–8%, MSH2 mutations for 1.5–3.0%, and *BRCA1* mutations for 1.0–2.5% [9,10,11,12].

The International Stand Up to Cancer (SU2C) Registry, which analyzed mCRPC cases, reported a similar distribution of somatic DDR mutations, with an overall incidence of 23%. Among these, *BRCA2* mutations were the most frequent at 13%, followed by *ATM* mutations at 7.3%, MSH2 mutations at 2%, and *BRCA1*/RAD51 mutations at 0.3% [12].

In the Profound study [9], which examined 2792 biopsies from mCRPC patients, HRR pathogenic variants were identified in 28% of cases, with no significant difference in frequency between the primary tumors (27%) and metastatic sites (32%). These findings raise important questions regarding the clinical implications of HRR gene mutations in the management of PCa. A similar pattern was observed in a study of 316 Chinese patients (including both mPCa and high-risk localized PCa cases), where 9.8% of patients had pathogenic mutations in DDR genes (95% confidence interval [CI]: 6.5–13%). Among these, 6.3% carried *BRCA2* mutations, 0.63% *BRCA1*, 0.63% *ATM*, and 2.5% had mutations in 15 other DDR-related genes. These data suggest that DDR gene mutations are relatively common and occur independently of ethnicity [10].

### 2.2. Role in Non-Metastatic PCa

Given the potential impact of DDR gene mutations on the oncological outcomes of treatments for localized PCa, including surgery, radiotherapy, and active surveillance, it is important to consider how somatic DDR mutations might influence the treatment efficacy. There is an urgent need for data on the implications of both germline and somatic DDR defects in early-stage PCa to better tailor treatment strategies from a precision medicine perspective [2,3].

In this context, the IMPACT study explored the role of Prostate-Specific Antigen (PSA) screening in men with BRCA mutations compared to controls, with a follow-up period of 3 years [13]. The study found that the differences between BRCA mutation carriers and non-carriers were consistently more pronounced for *BRCA2* mutations than for *BRCA1* mutations. The incidence of PCa detected at biopsy was 5.2% in *BRCA2* carriers compared to 3.0% in non-carriers. The distribution of high- or intermediate-risk PCa cases also differed significantly: 77% of *BRCA2* carriers were classified as high- or intermediate-risk compared to 40% of non-carriers. Additionally, the proportion of patients with International Society of Urological Pathology (ISUP) Grade 1 (low-risk) disease changed from 73% in non-carriers to 37% in *BRCA2* carriers. The IMPACT study [13] suggests that the analysis of *BRCA2* PVs associated with a total PSA measurement can increase the accuracy of a PCa screening program, reducing the risk of overdiagnosis for non-clinically significant tumors. These findings also suggest that BRCA mutation carriers may have a more aggressive form of disease, which should be considered when selecting treatment options.

In line with this, Chen RC and Carter HB et al. investigated whether germline mutations were associated with grade reclassification (GR) in patients undergoing active surveillance (AS) for PCa [14,15]. Analyzing 1211 PCa patients enrolled in AS, the study found that the incidence of tumor staging upgrades at 2, 5, and 10 years was higher in *BRCA2* carriers than in non-carriers: 27%, 50%, and 78% for *BRCA2* carriers versus 10%, 22%, and 40% for non-carriers. However, the retrospective nature of these studies, as well as the lack of image-guided biopsies and multiparametric magnetic resonance imaging (mMRI), limits the ability to draw definitive conclusions [14,15]. Further prospective studies are needed to better evaluate the role of AS in patients with BRCA germline mutations.

The prognostic significance of HRR gene mutations in localized disease has also been explored in the context of active treatments, such as surgery and radiotherapy. Castro et al. examined the impact of BRCA mutations on metastatic relapse and cause-specific survival (CSS) following radical treatments (surgery and radiotherapy) for localized PCa [16]. At 3, 5, and 10 years after treatment, 97%, 94%, and 84% of non-carriers were free from metastasis compared to 90%, 72%, and 50% of BRCA mutation carriers (*p* < 0.001). The CSS rates were significantly higher in non-carriers at 3, 5, and 10 years (99%, 97%, and 85%, respectively) compared to carriers (96%, 76%, and 61%, respectively; *p* < 0.001). A multivariate analysis confirmed that BRCA mutations were an independent prognostic factor for both metastasis-free survival (MFS) (hazard ratio [HR]: 2.36; 95% confidence interval [CI], 1.38–4.03; *p* = 0.002) and CSS (HR: 2.17; 95% CI, 1.16–4.07; *p* = 0.016) in non-metastatic PCa patients submitted to surgery or radiotherapy [16].

A similar study by Martinez Chanza et al. in a retrospective cohort of 380 patients with localized hormone-sensitive PCa found that BRCA mutations were associated with a higher risk of relapse after primary treatment [17]. Furthermore, Marshall et al. conducted a retrospective analysis on PCa patients, which revealed that *BRCA1/2* mutations were more frequently associated with high-grade disease (Gleason score ≥8, *p* = 0.00003), advanced tumor stage (T3/T4, *p* = 0.003), nodal involvement (*p* = 0.00005), and metastases at diagnosis (*p* = 0.005) compared to non-carriers [18]. Several data in the literature support the possible role of HRR PV analyses in the non-metastatic setting of PCa [13,14,15,16,17,18,19,20]; however, prospective studies are necessary to arrive at a guideline recommendation [8].

### 2.3. Role in Metastatic PC

Several studies have explored the prognostic significance of DDR mutations in mCRPC patients treated with standard therapies. In a retrospective study of 319 mCRPC patients, Annala et al. found that carriers of DDR mutations had significantly shorter progression-free survival (PFS) compared to non-carriers (3.3 vs. 6.2 months, *p* = 0.01) when treated with first-line androgen receptor pathway inhibitors (ARPIs) [21]. In contrast, Antonarakis et al., in a study of 172 mCRPC patients receiving first-line ARPIs, observed a trend toward longer PFS in *ATM* and *BRCA1/2* mutation carriers compared to non-carriers (15 vs. 10.8 months; *p* = 0.090) [22].

The Prorepair-B trial was the first prospective study to evaluate the prognostic impact of *BRCA1/2* and other DDR gene mutations on cause-specific survival (CSS) in mCRPC patients [1]. The study, which analyzed all DDR mutations collectively, did not find significant differences in CSS between carriers and non-carriers (DDR mutation carriers 23.3 months vs. non-carriers 33.2 months; *p* = 0.264; HR: 1.32; 95% CI: 0.81–2.17). However, when looking specifically at *BRCA2* mutations, the study found that *BRCA2* mutation carriers had significantly worse CSS compared to non-carriers (*BRCA2* carriers 17.4 months vs. non-carriers 33.2 months; *p* = 0.027; HR: 2.10; 95% CI: 1.07–4.10). Further subgroup analysis revealed that *BRCA2* mutation carriers had a shorter CSS when treated with a docetaxel-ARSI sequence (10.7 months) compared to those treated with an ARPI-docetaxel sequence (24.0 months) [1]. These findings suggest that *BRCA2* mutations have a negative impact on the outcomes of mCRPC and that the choice of first-line therapy may influence the prognosis for *BRCA2* mutation carriers [1].

In the setting of mCRPC, the evidence for a prognostic and decision-making role on the therapeutic choice based on the presence of HRR PVs is supported by prospective studies, so as to arrive at a guideline recommendation [8]. However, to date, the data are particularly concentrated on the analysis of BRCA and limitedly explore other HRR mutations.

## 3. How to Detect

DDR alterations can arise early in the development of aggressive tumors and may be detectable in prostate biopsies or prostatectomies. However, other alterations may emerge later during disease progression, in which case metastatic tumor biopsies might be the preferred method for detection.

### 3.1. Somatic Samples: Recent vs. Archived

In the Profound study, a total of 4858 tissue samples were centrally tested and analyzed for HRR PVs. Next-Generation Sequencing (NGS) results were successfully obtained for 58% of the samples, representing 69% of the patients. Of the total samples collected, 83% were from primary tumors (96% of which were archival and 4% were newly obtained), and 17% were from metastatic tumors (67% archival and 33% newly obtained). The NGS results were more commonly obtained from newly acquired samples (63.9%) compared to archived samples (56.9%) and from metastatic samples (63.9%) compared to primary tumor samples (56.2%) [9]. The authors noted that DNA yield had a greater impact on the success of the NGS results (AUC = 0.6292) than the other variables, such as sample age or tumor content percentage. Although the NGS success rates decreased as the sample age increased, approximately 50% of the samples older than 10 years still produced viable results. Another factor that affected eligibility was the total tissue volume. In the Profound study, the samples with a tissue volume greater than 0.6 mm^3^ had a higher rate of generating NGS results (58.9%; 95% CI, 57.4–60.3) compared to samples with volumes of 0.2 to ≤0.6 mm^3^ (32.1%; 95% CI, 24.4–40.6) [8]. Overall, the Profound study demonstrated that tissue testing for HRR alterations is feasible, with high-quality tumor tissues being crucial for generating NGS results [9].

The analysis of DDR mutations in PCa should prioritize somatic testing before germline testing. When DDR gene mutations arise during disease progression, metastatic tumor biopsies are the ideal method for identifying these molecular alterations. However, obtaining biopsies from metastatic lesions can be challenging or even unfeasible due to their location and the invasive nature of the procedure. Additionally, a single biopsy may not capture the full tumor heterogeneity across the different metastases. Furthermore, processing bone biopsy samples, which typically requires decalcification, can lead to reduced DNA yield and quality [1,2,3]. In the Profound study, the bone samples showed a lower proportion of successful NGS results compared to other biopsy sites (42.6%) [9]. However, recent preparation techniques with decalcification are also improving results in bone samples.

### 3.2. Germline Analysis in PC

A germline analysis is closely linked to genetic counseling for families of PCa patients with somatic pathogenic variants (PVs). Pre-test counseling should gather detailed family history (spanning at least three generations) and discuss available genetic testing options. Post-test counseling should focus on interpreting the results, explaining the potential risks for various cancers, and outlining the need for any intensive screening strategies. Germline testing can be performed using various approaches, such as small-focused panels (5–6 genes), cancer-specific panels (10–15 genes), or large comprehensive panels (≥80 genes), with tests increasing in complexity depending on the number of genes assessed. If a test result is negative or identifies a “variant of uncertain significance” (VUS), this should not be considered a pathogenic variant. VUS results are often eventually classified as non-pathogenic after further investigation [1,2,3].

Currently, BRCA testing for germline pathogenic variants is typically performed using validated methods, including Next-Generation Sequencing (NGS), and the results are generally confirmed through techniques such as Multiplex Ligation Probe Amplification (MLPA) or Multiplex Amplicon Quantification (MAQ) [3,4].

Although several methods exist for classifying constitutional BRCA variants, it is recommended to follow the criteria established by the Evidence-Based Network for the Interpretation of Germline Mutant Alleles (ENIGMA) [1,2,3,4]. The ENIGMA criteria are used to assess the clinical significance of sequence variants in genes related to breast, ovarian, and prostate cancer. ENIGMA has developed variant classification criteria that utilize both quantitative and qualitative (rule-based) methods to assess the clinical significance of PVs. Quantitative classifications are derived from multifactorial likelihood models that combine multiple lines of clinical data with the assumption that each feature is an independent predictor of PVs. This information is combined with the probability of pathogenicity based on bioinformatic predictors of variant effects on protein sequence or messenger RNA splicing, which are probabilities that have been calibrated against clinical information. In relation to BRCA, most variants considered pathogenic are premature truncation variants (including nonsense or frameshift) [3,4]. According to ENIGMA, the test results are classified into five categories based on the probability of pathogenicity (PP):Benign variants (PP < 0.001);Likely benign or of limited clinical significance (PP: 0.001–0.049);Uncertain significance (PP: 0.05–0.949);Likely pathogenic (PP: 0.95–0.99);Pathogenic (PP > 0.99).

### 3.3. Circulating DNA (cDNA)

The analysis of free circulating DNA (cDNA) presents a promising approach, especially in cases where obtaining tissue samples is difficult or not possible [23]. However, current data do not yet completely support the reliable use of cDNA testing in clinical practice. The first study to investigate cDNA in this context was the GHALAND study, a Phase 2 trial evaluating treatment outcomes with Niraparib in mCRPC patients with DDR mutations. The treatment efficacy was assessed based on the presence of circulating tumor cells (CTCs) starting from the eighth week of treatment. The best results were seen in the BRCA cohort, which had a 24% CTC response compared to the non-BRCA cohort [24]. However, the analysis of HRR mutations through a liquid biopsy and the determination of circulating DNA represent the potentially most interesting method capable of simplifying the difficulties of somatic determination [25].

Despite this promising initial result, commercial tests for cDNA analysis still show significant inconsistencies, with discrepancies as high as 40% between different tests [25].

## 4. Purpose for Experimental Design: Personal Experience

*Study protocol*: We developed an experimental multidisciplinary prospective study in patients with a new diagnosis of high-risk prostate adenocarcinoma at biopsy. The protocol has been approved by our internal committee at Sapienza University (000019_22_ATENEO2022_DDG N.n.4994/2022).

*The primary objective* was to evaluate and compare the tissue expression of pathogenic variants (PVs) of different genes involved in homologous recombination (HRDR) in patients with a first diagnosis of PCa (de novo) in relation to a metastatic or non-metastatic stage, tumor aggressiveness, and early risk of progression.

*Population:* de novo histological diagnosis of prostatic adenocarcinoma at biopsy was performed. A total of 22 consecutive patients who underwent prostate biopsy in our department and histological examination results with intermediate- or high-risk prostatic adenocarcinoma were included.

The inclusion criteria were as follows: patients with newly diagnosed, histologically confirmed prostate adenocarcinoma at biopsy (de novo) at intermediate or high risk (ISUP score 3–5, presence of Gleason Grades 4–5); stage metastatic or non-metastatic.

The exclusion criteria were as follows: current or previous androgen deprivation therapies, chemotherapy, radiotherapy, or other therapies capable of influencing the growth and progression of PCa.

Following the subsequent radiological staging, the cases were classified as non-metastatic (M0) or metastatic (M1) PCa.

### 4.1. Methods

#### 4.1.1. Urologic Evaluation

All cases were submitted for local staging with multiparametric magnetic resonance (mMR) and targeted biopsy. Systemic staging to classify between metastatic and non-metastatic cases was obtained through a Positron Emission Tomography, Computed Tomography (PET CT) scan (PSMA or choline). A family analysis up to the third generation for prostate, ovarian, breast, pancreatic, and colon cancers was reported (positive familiarity described as one or more first- or second-degree relatives), and longitudinal assessments for early (within 12 months) biochemical (total PSA) and/or radiological progression were analyzed. Table 1 and Table 2 describe the patient characteristics and results from the genetic analysis.

#### 4.1.2. Pathologic Preparation

In each case, a selection of the most representative prostate adenocarcinoma tissue with Grades 4–5 at biopsy and a description of cellularity for each slide were performed. In some cases, only one slice was obtained and named “A”. In other cases, up to three representative slices were prepared and named “A”, ”B”, and ”C” without a specific order or classification. At present, formalin-fixed paraffin-embedded (FFPE) tumor samples from 22 PCa patients have been collected. The diagnosis of cancer samples was evaluated by a genitourinary pathologist on hematoxylin and eosin (H&E)-stained slides. From each patient, we had chosen from one to three slides. The selected samples were cut into 5–10 × 5 µm sections with the last H&E-stained 4 µm sections to confirm and assess tumor cellularity.

#### 4.1.3. Genetic Analysis

DNA was extracted from 43 formalin-fixed paraffin-embedded (FFPE) slices obtained from the prostate adenocarcinoma biopsies of 22 patients using the MagCore^®^ Super Automated Nucleic Acid Extractor (Diatech Lab Line, Jesi, Italy) by the MagCore^®^ Genomic DNA FFPE One-Step Kit according to the manufacturer’s instructions. The process started with the preparation of Proteinase K, which was reconstituted in a buffer solution and stored at −20 °C until needed. The tissue samples were placed in a cartridge well along with Sula Oil and Proteinase K, allowing the system’s one-step heating method to simultaneously melt the paraffin and lyse the tissue. In the final step, genomic DNA is purified using cellulose-coated magnetic bead technology. The DNA concentration was assessed using the fluorometric method using Qubit 3.0 and the Qubit™ 1X dsDNA High Sensitivity (HS) (ThermoFisher, Waltham, MA USA). The DNA quality was tested by a quantitative PCR using the Archer PreSeq DNA QC Assay (Archer, Integrated DNA Technologies, Boulder, CO, USA) on the Archer VariantPlex instrument (Boulder, CO, USA).

For the Next-Generation Sequencing (NGS) results, the DNA libraries were prepared by an amplicon-based protocol using the Archer™ VARIANTPlex™ Pan Solid Tumor Panel (Boulder, CO, USA) according to the manufacturer’s instructions. The quality and concentration of the prepared libraries were evaluated using Qubit 3.0 and the Qubit™ 1X dsDNA High Sensitivity (HS) and TapeStation with D1000 ScreenTape. The libraries that met the quality threshold were pooled and run on the NextSeq 550dx. The resulting FASTQs were uploaded to the Archer Server (Boulder, CO, USA) for analysis and variant reporting. An alignment of the reads over the reference genome and variant calling were then performed by the ArcherDX analysis software version 3.2 (Figure 1).

The Archer PreSeq DNA QC Assay utilizes quantitative PCR (qPCR) to identify DNA samples of sufficient quality to produce an adequate sequencing library using Archer VARIANTPlex assays. The Archer DNA QC Assay is used to evaluate the quantity of amplifiable DNA in a sample in relation to a known assay standard. This SYBR^®^ Green assay amplifies a 100 bp genomic DNA sequence in the sample and the assay standard in two separate reactions. A comparison of the two resultant quantification cycle (Cq) values results in a ΔCq or DNA QC Score. The DNA QC Score holds predictive value for library yield and can be used to gauge the amount of input material required for successful Archer VARIANTPlex library preparation. The DNA QC Score obtained is a direct measurement of functional DNA templates, which provides an indicator of library yield and library quality metrics when using Archer VARIANTPlex assays.

Somatic variants were classified according to the standards and guidelines of the American Society of Clinical Oncology, the joint recommendations of the Clinical Genome Resource (ClinGen, Bethesda, MD, USA), the Cancer Genomics Consortium (CGC, Lafayette, LA, USA), and the Variant Interpretation for Cancer Consortium (VICC, Toronto, Ontario, Canada) using the ArcherDX analysis software version 3.2 (Boulder, CO, USA) [26]. The DNA variants were indeed categorized in four different classes according to their clinical impact as follows:

Tier I represents variants with strong clinical significance (biomarkers that predict response or resistance to FDA-approved therapies for a specific type of tumor, biomarkers included in professional guidelines that predict response or resistance to therapies for a specific type of tumor, and biomarkers that predict response or resistance to therapies for a specific type of tumor based on well-powered studies with consensus from experts in the field).

Tier II represents variants with potential clinical significance (biomarkers that predict response or resistance to therapies approved by the FDA or professional societies for a different type of tumor, biomarkers that serve as inclusion criteria for clinical trials, and biomarkers that show plausible therapeutic significance based on preclinical studies).

Tier III represents variants with unknown clinical significance.

Tier IV represents variants that are benign or likely benign.

The classification is performed using oncogenicity points, which are ranked on a scale to determine the class of the variant under evaluation. These scores will be the result of a combination of population data, functional data, predictive data, the presence of the mutation in a cancer hotspot, and computational evidence [27].

Only variants assigned to Tier I (oncogenic), Tier II (likely oncogenic), and Tier III (variant of uncertain significance) were included in this study.

### 4.2. Findings

Among the DNA samples from the 43 FFPE slices, the concentration and the quality of 25 were sufficient to proceed with the preparation of the NGS libraries (Table 1). Thereafter, the sequencing of seven samples failed due to low library concentrations (Table 2).

**Table 2 biology-14-00117-t002:** Clinical characteristics of the PCa cases who failed sequencing. Cellularity on histological slices. Concentration and quality of DNA extraction. NGS analysis.

Patient	Age (Years)	Familiarity	Aggressiveness	Early Progression	TNM Staging System	Years from Sample Collection	Slice	Cellularity	Concentration of the Extracted DNA (ng/uL)	Quality of the Extracted DNA	NGS Library Concentration (nMol)
				(12 Months)							
1	58	no	ISUP 4	yes	M1	1	A	60%	3.9	Low	Not performed
							B	60%	6.4	Low	Not performed
7	69	no	ISUP 3	yes	M1	1	A	70%	10.6	Medium	Low
							B	60%	11.9	Medium	Low
9	65	no	ISUP 4	yes	M1	0	A	40%	11.5	Medium	Low
10	78	no	ISUP 5	yes	M1	0	A	80%	6.2	Medium	Low
11	67	no	ISUP 3	yes	M0	1	A	40%	7.8	Medium	Low
							B	50%	90.0	Low	Not performed
							C	50%	4.0	Low	Not performed
12	54	no	ISUP 3	no	M0	4	A	30%	3.9	Low	Not performed
							B	20%	5.5	Medium	Low
15	66	no	ISUP 3	yes	M0	2	A	40%	75.6	Medium	Low
							B	60%	6.9	Medium	Low
17	66	yes (1 brother)	ISUP 3	yes	M0	1	A	20%	4.5	Low	Not performed
							B	40%	3.8	Low	Not performed
18	78	no	ISUP 4	yes	M1	0	A	80%	4.2	Low	Not performed
22	76	no	ISUP 4	yes	M1	1	A	40%	1.8	Low	Not performed
							B	50%	0.2	Low	Not performed

Therefore, the genetic NGS data were obtained for 17 samples belonging to 12 different patients. The DNA variants were filtered by the following criteria: (i) read depth >300X; (ii) MAF (minor allele frequency) in the gnomAD global population database <5%; (iii) number of times a variant was observed on the system in different runs <5; and (iv) variant allele frequency (VAF) >2%. The analysis software calculates the VAF as the ratio of the number of reads for the mutant allele to the total number of reads present for the specific locus. Only reads meeting the analysis software’s quality parameters are counted.

In total, 14 different DNA variants were prioritized in 10 bioptic samples from 9 different patients. The variant allele frequency (VAF) was between 3% and 53%. Since blood DNA samples from the patients were not available, we cannot exclude that the variants with a high VAF were constitutional (Table 1).

Five samples from the non-metastatic patients (Cases 3, 6, 8, 14, and 21) and one sample from a metastatic case at first diagnosis (Case 20), all with early progression (no later than 12 months from treatment) (except Case 3 without early progression), were found to carry at least one Tier I/II variant.

According to the standards and guidelines of the American Society of Clinical Oncology, the variant NM_000314.8:c.955_958del (p.Thr319Ter) in the *PTEN* gene was classified as Tier I. Indeed, the Thr319Ter has been validated for an FDA-approved therapy for the treatment of patients with *PTEN*-mutant ER+/HER2 metastatic breast cancer, but the clinical benefit of this combination in patients with *PTEN* Thr319Ter-mutant prostate cancer is unknown. This variant was only found in one of the two FFPE samples from Patient 21, a patient with familiarity (one brother with prostate cancer), aged 76 years, non-metastatic PCa, and ISUP 3, with early biochemical progression in a hormone-sensitive tumor.

Three out of the fourteen different prioritized variants were found in the *ATM* gene, all non-metastatic cases with early progression. A *BRCA1* oncogenic variant was detected in only one PCa case (Case 6), non-metastatic, early progression, and familial for prostate cancer. The same patient also carried one Tier II variant in *KMT2D*, as well as a variant of uncertain significance in *RB1*. Very recently, in a mouse, the Kmt2c deficiency has been found as a metastatic driver [28]. A truncating Tier II variant in the *CDH1* gene was found in the PCa biopsy from Patient 20 (metastatic and early progression); the role of *CDH1* mutations in non-gastric cancers has yet to be defined. *BRCA2* variants were not identified. Of note, the PCa biopsy of two patients (Cases 8 and 21, both non-metastatic and early progression) carried variants in *FOXA1*, a gene that plays an important role in regulating steroid receptor functions and whose expression has been reported to be related to several human cancers as both an oncogene and a tumor suppressor gene, depending on the specific cancer types of the gene [29]. In particular, *FOXA1* is thought to play an important role in prostate cancer progression and may be a factor leading to aggressive prostate cancer [30].

Five samples passed the QC metric for MSI (microsatellite site instability), and no instability was detected. No copy number variants (CNVs) were identified in any sample.

### 4.3. Discussion

To date, the analysis of HRR PVs in PCa patients remains complex, time-consuming, and at high cost, so that it is often limited to a single gene, such as BRCA. BRCA PVs have been shown to have the highest predictive value in patients with metastatic PCa regarding sensitivity for PARP inhibitors therapy. However, the predictive value of a more extensive analysis of a large pool of PVs in the HRR family could offer prognostic indications and responses to therapies not only limited to the use of PARP inhibitors. This expansion of analysis could be particularly useful in an earlier phase of the disease in non-metastatic PCa. The high complexity and costs of this type of analysis can limit its use in clinical practice, and a selection in high-risk patients or based on family histories is certainly necessary. It is possible that HRR genes other than BRCA and other PVs may be predominant and prognostically useful in this earlier disease setting. To determine their clinical impact, somatic variants have been classified according to the standards and guidelines of the American Society of Clinical Oncology, the joint recommendations of the Clinical Genome Resource (ClinGen), the Cancer Genomics Consortium (CGC), and the Variant Interpretation for Cancer Consortium (VICC) and categorized into four different classes [26]: Tier I, variants with strong clinical significance (level A and B evidence); Tier II, variants with potential clinical significance (level C or D evidence); Tier III, variants with unknown clinical significance; and Tier IV, variants that are benign or likely benign.

Our personal experience is certainly limited by the complexity and costs of an analysis extended to a large panel of HRR genes. The limited number of cases examined does not allow for a comparison of the results based on non-metastatic versus metastatic staging. However, in the non-metastatic high-risk PCa population, different Tier I-III variants have been isolated, such as *PTEN*, *ATM*, *BRCA1*, *KMT2D*, *RB1*, and *FOXA1*.

*FOXA1*, a forkhead protein, is a partner of the estrogen (ER) and androgen (AR) receptors and is involved in both breast and prostate cancer pathogenesis [31]. Progress has been made in the understanding of how *FOXA1* influences nuclear AR and ER activity. *FOXA1* resulted in mutations in 1.8% of breast and 3–5% of prostate cancers, and there is evidence of somatic changes that influence the DNA sequence under *FOXA1*-binding regions and its interaction with AR and ER in tumor cells [31]. In a series of PCa cases, Grasso et al. discovered the *FOXA1* mutation both in non-metastatic or CRPC cases, and the mutation in the coding sequence occurred either in or around the forkhead DNA-binding domain or in the C-terminal transactivation domain [31]. These mutations affect *FOXA1*’s affinity for its canonic binding site, potentially resulting in binding a new genomic location with effects on the transcriptional program within PCa.

The *RB1* gene is the first tumor suppressor gene identified, whose mutation and inactivation is the cause of a human cancer, retinoblastoma, in pediatric age [32]. *RB1* encodes protein pRb, which dynamically regulates the location-specific assembly of protein complexes on DNA in response to the output of various signaling pathways [32]. Different findings suggest that the *RB1* mutation cannot contribute to the initiation of tumorigenesis, but it is clearly a late event in tumor progression [32]. However, in PCa, the loss of *RB1* is an early event, and it is related to PCa development, progression, and treatment resistance [33] (Table 3).

Histone methyltransferase *KMT2D* plays a critical role as a human oncogene in PCa; its deficiency inhibits tumor cell proliferation, inducing apoptosis, and a possible association with Interleukine-6 (IL-6) has been hypothesized [34]. The fact that some cytokines act as a link between *KMT2D* and the tumor microenvironment is of particular interest. *KMT2D* oncogene expression in PCa has also been related to oxidative gene expression and as a mediator of the cellular response to oxidative stress [34].

*ATM* mutations have been included in a list of actionable mutations for PARP inhibitor trials; however, PARPi has shown minimal clinical activity in *ATM* PVs in prostate cancer [6,22]. The ataxia telangiectasia mutated (*ATM*) gene is one of the largest genes in the genome and is a tumor suppressor gene that encodes PI3K-related serine/threonine protein kinase, which identifies DNA damage [35]. Among metastatic PCa, the frequency of *ATM* PVs is reported as 7.0%, whereas among non-metastatic PCa, it is reported as 1.5% [9,10,11,12]. An analysis of *ATM* variants in metastatic PCa showed no clear correlation with survival or treatment response [35].

*PTEN* tumor suppressor gene mutations are among the most common in PCa, and inactivation of *PTEN* by deletion or mutation is identified in approximately 20% of non-metastatic PCa samples after surgery and in 50% of castration-resistant metastatic PCa [36]. Loss of *PTEN* function leads to activation of the PI3K-AKT pathway and is strongly associated with negative oncological outcomes and PCa progression [36]. Among prognostic DNA biomarkers, *PTEN* loss is one of the most promising in PCa. *PTEN* inactivation in PCa is associated with an increased risk for high grade and stage, short-term biochemical and radiological progression after primary treatment, and metastatic and castration-resistant progression [36]. Moreover, *PTEN* loss has been associated with decreased response to ARPI in metastatic CRPC cases [36]. However, the most exciting context for *PTEN* variant analysis as a predictor of therapeutic response has been in the setting of targeted therapies to PI3K/AKT/mTOR signaling [36].

## 5. Conclusions

Prostate cancer is one of the most common cancers affecting men, with a substantial impact on both survival and quality of life, as well as significant social and psychological consequences. Despite extensive research in recent years, awareness of the genetic risks associated with PCa remains low. There is a critical unmet need to develop effective health promotion strategies and screening programs targeted at younger men with genetic mutations linked to a higher risk of developing PCa.

Currently, there is no international consensus on the optimal management approach for men at genetic risk for prostate cancer. However, some key findings have emerged:The incidence of pathogenic variants (PVs) in HRR genes among men with metastatic PCa ranges from 11% to 33%, which is notably higher than in non-metastatic prostate cancer (nmPC). Within the metastatic setting, *BRCA2* mutations are more prevalent compared to other HRR gene mutations.Identifying somatic or germline HRR PVs, particularly *BRCA2* mutations, plays a crucial role in personalizing treatment with PARP inhibitors in metastatic castration-resistant prostate cancer (mCRPC). This approach has shown significant improvements in radiographic progression-free survival (rPFS) and overall survival (OS). As a result, this strategy has been recommended by international guidelines and has received approval from both the FDA and EMA.

Several points remain undefined, and relevant unmet needs must be addressed:-To better define clinical and pathological characteristics of newly diagnosed prostate cancer associated with DDR gene defects.-To offer a platform for approaching personalized medicine based on the genetic assessment of PVs in DDR genes in a non-metastatic stage.-To define whether the expression of PVs of DDR genes is also relevant in non-metastatic prostate cancer at high risk.-To define whether *BRCA2* remains the main PV expressed also in non-metastatic prostate cancer cases or other PVs for different DDR genes are similarly expressed and useful.-To simplify the detection of PVs of DDR genes, exploring not only the somatic but also other approaches.

## Figures and Tables

**Figure 1 biology-14-00117-f001:**
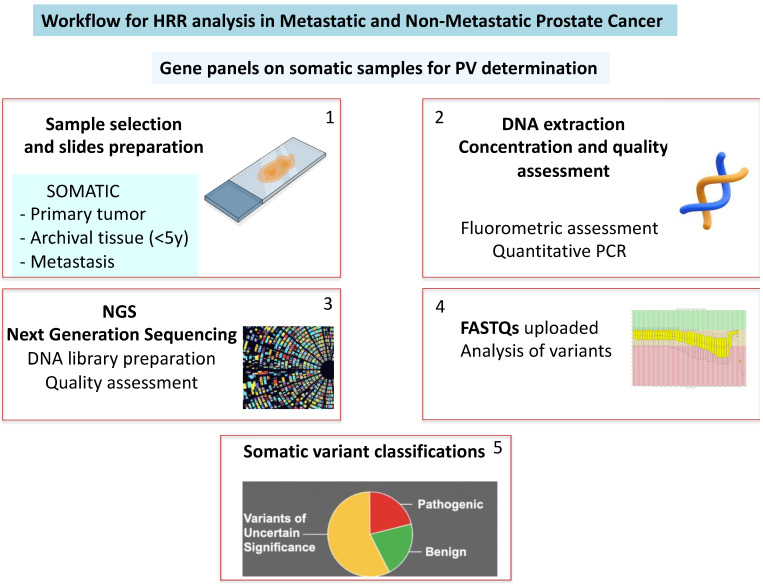
Workflow for HRR analysis and pathogenic variants (PVs) determination.

**Table 1 biology-14-00117-t001:** Clinical characteristics of the successfully analyzed PCa cases. Cellularity on histological slices. Concentration and quality of DNA extraction. NGS analysis and genetic results.

Patients	Age (Years)	Familiarity	Aggressiveness	Early Progression	TNM Staging System	Years from Sample Collection	Slice	Cellularity	Concentration of the Extracted DNA (ng/ul)	Quality of the Extracted DNA	NGS Libraries’ Concentration (nMol)	Genetic Results				
				(12 Months)								Gene	Variant	Classification	Variant Allele Frequency (%)	Microsatellites Instability
2	65	no	ISUP 4	yes	M1	1	A	80%	12.0	Medium	Low	NGS failed				
							B	70%	93.6	Medium	46.5	*DDR2*	NM_006182.4:c.106A > C, p.Met36Leu	TIER III	52.0%	no
3	59	no	ISUP 3	no	M0	1	A	40%	61.0	Medium	91.3	*ATM*	NM_000051.4:c.3044_3045delinsCCTT, p.Gln1015Profs	TIER I	47.0%	no
							B	50%	52.4	Medium	128.5	*ATM*	NM_000051.4:c.3044_3045delinsCCTT, p.Gln1015Profs	TIER I	45.90%	no
4	66	no	ISUP 3	yes	M0	2	A	30%	58.8	Medium	81.3	wild type				no
5	78	no	ISUP 3	yes	M0	4	A	30%	5.54	Medium	Low	NGS failed				
							B	30%	7.36	Medium	36.3	wild type				no
6	68	yes (1 brother)	ISUP3	yes	M0	1	A	30%	27.4	Low	n.p.	NGS n.p.				
							B	60%	68.4	Medium	87.2	*RB1* *BRCA1* *KMT2D*	NM_000321.3:c.1976A > G, p.Tyr659Cys NM_007294.4:c.5150del, p.Phe1717SerfsNM_003482.4:c.9979C > T, p.Gln3327Ter	TIER III TIER I TIER II	50.1% 46.9% 2.8%	no
8	68	no	ISUP 5	yes	M0	1	A	35%	77.8	Medium	65.4	wild type				no
							B	40%	93.8	Medium	139.6	*ATM* *FOXA1*	NM_000051.4:c.2272_2301del, p.Glu758_Thr767del NM_004496.5:c.787del, p.Gln263ArgfsTer58	TIER III TIER III	6.4% 6.2%	no
							C	40%	6.83	Medium	Low	NGS failed				
13	66	no	ISUP 3	no	M0	2	A	30%	23.0	Medium	Low	NGS failed				
							B	40%	27.8	Medium	29.7	wild type				no
14	57	no	ISUP 5	yes	M0	0	A	70%	54.8	Medium	29.2	*ATM*	NM_000051.4:c.2093C > G, p.Ser698Ter	TIER I	55.0%	no
							B	80%	118	Medium	Low	NGS failed				
							C	70%	76.6	Medium	Low	NGS failed				
16	57	no	ISUP 4	no	M0	2	A	80%	38.4	Medium	Low	NGS failed				
							B	80%	93.6	Medium	36.4	wild type				no
19	64	no	ISUP 4	no	M0	0	A	75%	7.98	Medium	46.9	wild type				no
							B	60%	3.92	Medium	28.6	wild type				no
20	65	no	ISUP 5	yes	M1	1	A	70%	13.5	Medium	93.2	*CDH1* *FANCA* *PLCB4*	NM_004360.5:c.1912dup, p.Trp638Leufs NM_000135.4:c.11C > G, p.Ser4Trp NM_000933.4:c.713C > T, p.Thr238Met	TIER II TIER III TIER III	12.4% 46.0% 43.2%	no
							B	70%	7.14	Medium	76.5	*CDH1* *FANCA* *PLCB4*	NM_004360.5:c.1912dup, p.Trp638Leufs NM_000135.4:c.11C > G, p.Ser4Trp NM_000933.4:c.713C > T, p.Thr238Met	TIER II TIER III TIER III	10.5% 44.6% 42.9%	no
21	76	yes (1 brother)	ISUP 3	yes	M0	1	A	50%	11.4	Medium	44.6	*FOXA1* *PTCH1*	NM_004496.5:c.806A > T, p.Glu269Val NM_000264.5:c.2492A > G, p.Tyr831Cys	TIER III TIER III	50.5% 53.5%	no
							B	40%	11.9	Medium	63.7	*FOXA1* *PTCH1* *PTEN*	NM_004496.5:c.806A > T, p.Glu269Val NM_000264.5:c.2492A > G, p.Tyr831Cys NM_000314.8:c.955_958del, p.Thr319Ter	TIER III TIER III TIER I	45.4% 46.2% 4.8%	no

**Table 3 biology-14-00117-t003:** Definition, control activity, and incidence from the literature of PVs in DDR genes detected in our experience.

Pathogenetic Variants DDR Genes	Definition	Control Activity	Frequency in Prostate Cancer from the Literature
*FOXA1* [31]	Forkhead box A1 gene	Partner of androgen and estrogen receptors	3–5%
*RB1* [32]	Retinoblastoma gene1	Tumor suppressor gene encoding protein pRb	5–20%
*KMT2D* [34]	Histone-lysine N-methyltransferase 2D gene	Oncogene influencing tumor cell proliferation. Possible association with Interleukine-6 and microenvironment activity	5–8%
*ATM* [35]	Ataxia telengiectasia mutated gene	Tumor suppressor gene encoding PI3K-related serine/threonine protein kinase. Identification of DNA damage	1.5–7%
*PTEN* [36]	Phosphatase and tensin homolog gene	Tumor suppressor gene related to the PI3K-AKT pathway	20–50%

## Data Availability

Experimental data are available and all presented in Table 1 and Table 2.

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
