# Peer review of "Early Detection of the Pathogenetic Variants of Homologous Recombination Repair Genes in Prostate Cancer: Critical Analysis and Experimental Design"

_biology, 2025, doi:10.3390/biology14020117_

Round 1
Reviewer 1 Report
Comments and Suggestions for Authors
The authors of this manuscript evaluated prostate cancer and its genetic basis. Overall, the manuscript is not well organized, and its cohesion and coherence are poor. Additionally, the aim of the study is not clearly articulated, and the type of article is not specified. Several comments that require further clarification, explanation, or modification are listed below:
- The authors used several abbreviations in the “Abstract” or “Simple Summary” sections that were not introduced in their full forms beforehand. Please revise these sections to address this.
- The “Simple Summary” section is unnecessary. All summary information should be included in the “Abstract” section.
- The type of study should be mentioned in the title. Why was the article classified as a “review”? Is it truly a review article?
- Please organize the manuscript into standard sections, including Introduction, Methods, Results, Discussion, and Conclusion.
- Please provide appropriate references for the information in the first paragraph.
- Abbreviations should be introduced in their full forms in their first appearance, followed by the abbreviated forms, afterwards. Please ensure this is applied throughout the manuscript.
- It is inappropriate to present information in bullet points in the Introduction section. Please revise this format.
- There are several instances in the manuscript where information is presented without proper references. This issue is particularly evident in the “Role in Non-Metastatic PC” section.
- Some data and information from the literature are included without organization. It appears as though they were simply copy-pasted, and at times, all numerical data from other studies are presented without clear justification. Please revise this to improve the structure and relevance of the information.
- Around the middle of the manuscript, the authors unexpectedly introduce an experiment that they conducted. This shift in focus, from reviewing the literature to presenting original research, disrupts the flow of the article. This approach lacks professionalism in terms of scientific writing.
- The limitations of the study should be discussed at the end of the Discussion section.
- Please consider a thorough English language revision to improve the manuscript's overall proficiency.
I wish you the best of luck and success in revising the manuscript.
Comments on the Quality of English Language- Please consider a thorough English language revision to improve the manuscript's overall proficiency.
Author Response
Dear Reviewer, thank you for your constructive comments and to give us the possibility to revise and improve our manuscript. We answered to your questions and now the manuscript improved. Modifications are evidenced in heavenly.
- The authors used several abbreviations in the “Abstract” or “Simple Summary” sections that were not introduced in their full forms beforehand. Please revise these sections to address this.
As requested all abbreviations in Abstract and Simple Abstract have been explained the first time
- The “Simple Summary” section is unnecessary. All summary information should be included in the “Abstract” section.
I can agree with you and I can remove the “Simple Summary” section. However this Section is requested from the Journal as a shorter Abstract for their use. I have removed, in case Editor and journal will request to replace we have ready.
- The type of study should be mentioned in the title. Why was the article classified as a “review”? Is it truly a review article?
As requested the type of article is reported in the Title page. This is an invited paper for a Special Issue. The proposal shared with the Editor of the Special Issue is to produce a Review article focused on the criticality of HRR analysis in an early stage of prostate cancer, also presenting a personal experimental design that we developed on a limited group of PCa cases. Therefore the organization of the manuscript is not the standard for an Original Article with Introduction, methods, results, Discussion. We started focusing on the topic of HRR analysis in PCa, clinical data in non metastatic setting, and after, in the middle of the manuscript we present our personal experimental design with methods and findings ( we now changed paragraph tile in “Purpose for Experimental Design).The aim is to suggest a possible organization and design to investigate the topic of HRR PVs in early disease and which methods and results could be expected ,But this part is not an Original trial inside a review article . The title of the manuscript describe this design: “ critical analysis and Experimental design”.
- Please organize the manuscript into standard sections, including Introduction, Methods, Results, Discussion, and Conclusion.
I agree with you that the organization of Paragraphs in our manuscript is in part unusual but in our opinion still able to clarify the flow of information given to the reader. This is an invited paper for a Special Issue. The proposal shared with the Editor of the Special Issue is to produce a Review article describing the criticality of HRR analysis in an early stage of prostate cancer, also presenting a personal experimental design on a limited group of PCa cases. The same organization has been recently shared with the Journal in the publication: Biology 2024, 13(12), 1024; https://doi.org/10.3390/biology13121024. Therefore the organization of the manuscript is not the standard for an Original Article with Introduction, methods, results, Discussion. We started focusing on the topic of HRR analysis in PCa, clinical data in non metastatic setting, and after,in the middle of the manuscript we present our personal experimental design with methods and findings ( we now changed paragraph tile in “Purpose for Experimental Design”). The aim is to suggest a possible organization and design to investigate the topic of HRR PVs in early disease and which methods and results could be expected ,But this part is not an Original trial inside a review article. As you requested we in part changed and respect the sequence you indicated with Introduction, Methods, Results, Discussion and Conclusions are now all present, but this is not an classic Original Paper.
- Please provide appropriate references for the information in the first paragraph.
As requested, references have been all revised and in particular in the first paragraph. New references have been enclosed (4,8,26,27,31-36).
- Abbreviations should be introduced in their full forms in their first appearance, followed by the abbreviated forms, afterwards. Please ensure this is applied throughout the manuscript.
We revised all abbreviation throughout the manuscript and now all are introduced in their full forms
- It is inappropriate to present information in bullet points in the Introduction section. Please revise this format.
As requested in Introduction the presentation of information in bullet points has been removed.
- There are several instances in the manuscript where information is presented without proper references. This issue is particularly evident in the “Role in Non-Metastatic PC” section.
Proper references have been enclosed in all manuscript and in particular in the section “ Role in Non-metastatic PC”. Each study is well referenced.
- Some data and information from the literature are included without organization. It appears as though they were simply copy-pasted, and at times, all numerical data from other studies are presented without clear justification. Please revise this to improve the structure and relevance of the information.
The entire manuscript has been revised and better organized so as to have a mor fluid path and each data has not been presented only as a number but explained in the meaning and context of the work.
- Around the middle of the manuscript, the authors unexpectedly introduce an experiment that they conducted. This shift in focus, from reviewing the literature to presenting original research, disrupts the flow of the article. This approach lacks professionalism in terms of scientific writing.
I agree with you that the organization of Paragraphs in our manuscript is in part unusual but in our opinion still able to clarify the flow of information given to the reader. This is an invited paper for a Special Issue. The proposal shared with the Editor of the Special Issue is to produce a Review article describing the criticality of HRR analysis in an early stage of prostate cancer, also presenting a personal experimental design on a limited group of PCa cases. The same organization has been recently shared with the Journal in the publication: Biology 2024, 13(12), 1024; https://doi.org/10.3390/biology13121024. Therefore the organization of the manuscript is not the standard for an Original Article with Introduction, methods, results, Discussion. We started focusing on the topic of HRR analysis in PCa, clinical data in non metastatic setting, and after,in the middle of the manuscript we present our personal experimental design with methods and findings ( we now changed paragraph tile in “Purpose for Experimental Design”). The aim is to suggest a possible organization and design to investigate the topic of HRR PVs in early disease and which methods and results could be expected ,But this part is not an Original trial inside a review article. As you requested we in part changed and respect the sequence you indicated with Introduction, Methods, Results, Discussion and Conclusions are now all present, but this is not an classic Original Paper
- The limitations of the study should be discussed at the end of the Discussion section.
As requested, limitations are discussed in the Discussion section 4.3.
- Please consider a thorough English language revision to improve the manuscript's overall proficiency.
An English language revision has been performed by native English speaking staff expert in scientific articles.
Reviewer 2 Report
Comments and Suggestions for Authors
The manuscript by Bottillo et al. reported an analysis of pathogenic variants of homologous recombination repair genes in prostate cancer patients. The authors summarized the impact of HRR pathogenic variants on the prostate cancer incidence, treatment efficiency. Then they reviewed the detection methods of DNA damage response mutations in prostate cancer patients. The authors selected 25 samples from 12 patients among 43 samples from 22 patients for next generation sequencing. The analysis results revealed oncogenetic and likely oncogenetic variants in ATM, BRCA1, PTEN, KMT2D and CDH1 genes. Variants of uncertain significance were found in ATM, DDR2, FANCA, FOXA1, PLCB4, PTCH1 and RB1 genes. Overall, I think their study provides valuable clinical data on HRR mutations in prostate cancer patients, which is now very important for improving the treatment efficiency in prostate cancer.
They are some questions need to be addressed.
1. The authors may want to introduce more on the concept of pathogenic variants, including how it is defined by the ENIGMA criteria in more detail.
2. Several abbreviations need proper explanation, including ISUP, PSA
Author Response
Dear Reviewer, thank you for your positive and constructive comments and to give us the possibility to revise and improve our manuscript. We answered to your questions and now the manuscript improved. Modifications are evidenced in red.
- The authors may want to introduce more on the concept of pathogenic variants, including how it is defined by the ENIGMA criteria in more detail.
The concept of pathogenic variants and HRD has been better described in paragraph 1 Background. Moreover, the ENIGMA criteria have been better described in paragraph 3.2. and reference 4 modified. The text was revised in paragraph 4.1.3. Genetic Analysis): we better explained the classification criteria of tumor-associated variants, without focusing only in BRCA1/2 classification ENIGMA criteria.
- Several abbreviations need proper explanation, including ISUP, PSA
All abbreviations along the text have been explained at their first appearance. In particular PSA in paragraph 2.2 and ISUP in paragraph 2.2
Reviewer 3 Report
Comments and Suggestions for Authors
Dear Authors.
Thank you for your research in this area, the study of pathogenic variants of HRR genes is really very important for the management of personalized therapy for prostate cancer. You have provided a great and high quality paper, but here are a few comments that I have noticed:
1) The manuscript contains several typographical errors that require correction. For example, on line 34, the word 'the' appears redundantly. On line 98, the correct abbreviations 'DDR' and 'PCa' are used instead of the typographical errors 'Ddr' and 'Pca'. Furthermore, the transition from the discussion of BRCA 1/2 genes on line 90 to "BCA" genes appears to be another typographical error. Additionally, "haematoxiline and eosine" on line 302 should be corrected to "haematoxilin and eosin," and "analyses" in Figure 1 should be amended to the singular form, "analysis."
2) Тhere is a discrepancy in the frequency data. There appears to be an inconsistency regarding the reported frequency of DDR gene alterations in non-metastatic prostate cancer (nmPCa). While the text states a range of 5-10% (lines 103-104), the most frequently observed pathogenic variant, BRCA2, is cited as having a frequency of 5.3%, which barely exceeds the lower bound of this range.
3) Table 1.
a) Column names may be given in full without a separate legend.
b) The meaning of the 'familiarity' column is not clear; if it is information about a close relative with a similar cancer type or mutation in the DDR gene, this should be stated explicitly. Related information appears only 2 pages later on line 354 without mentioning the type of cancer and/or mutation.
c) In the 'stage' column, 'metastatic' and 'non-metastatic' can be explicitly stated.
d) No explanation of what 'A', 'B', 'C' slices mean.
e) There is no explanation of what 'n.p.' means.
f) How the variants of allele frequency were estimated?
4) The ArcherDX analysis software has a reference to Figure 1 which does not explain what ArcherDX is.
5) Figure 1 shows a 'Workflow for HRR analysis in metastatic prostate cancer'. Is this workflow only suitable for mPCa? In Tables 1,2, samples from nmPCa patients were analysed using a different workflow?
6) Also, comments on the proposed workflow:
a) It seems to me that the quality of total genomic DNA isolation should be assessed by the presence of double-strand breaks by separating potential DNA fragments by gel electrophoresis. With qPCR, only the integrity of specific regions of the genome covered by primers can be assessed.
b) The 4th point is distributed between 3 and 5. NGS quality assessment is the quality control of fastq files. SNP variant analysis and classification is the final stage of SNP calling. If all the bioinformatic analysis is performed on a third party platform (such as ArcherDX) and not with your own tools, it is better to specify this explicitly.
7) There is no way to trace associations between PV DDR gene, cancer type and other parameters from Table 2. It is probably redundant here.
A few more comments on the paper as a whole:
The following considerations should be given when considering the overall manuscript:
1) A comprehensive discussion of DDR genes is required. While the manuscript mentions various DNA Damage Repair (DDR) genes, and Table 1 specifically lists genes associated with particular patient pathology variants, the discussion in the second paragraph should be expanded to explore the roles of other DDR genes (at least those detailed in the table). This discussion should include an evaluation of their specific contributions to pathogenesis and suggest potential reasons for their association with particular types of pathology, thereby providing a more thorough overview of the DDR pathway involvement in the study.
2) Justification of the Chosen Workflow: The presented workflow for HRR analysis does not appear to demonstrate a clear advantage over classical Sanger sequencing. It incorporates multiple comparable stages to those required for Sanger sequencing. Moreover, Sanger sequencing is a more cost-effective, accurate and time-efficient method for this application. Consequently, the manuscript should provide a more robust justification for the adoption of the chosen workflow, particularly in comparison to the established Sanger sequencing method, which may be more appropriate to the objectives of this study.
Author Response
Dear Reviewer, thank you for your positive and constructive comments and to give us the possibility to revise and improve our manuscript. We answered to your questions and now the manuscript improved. Modifications are evidenced in yellow.
- The manuscript contains several typographical errors that require correction. For example, on line 34, the word 'the' appears redundantly. On line 98, the correct abbreviations 'DDR' and 'PCa' are used instead of the typographical errors 'Ddr' and 'Pca'. Furthermore, the transition from the discussion of BRCA 1/2 genes on line 90 to "BCA" genes appears to be another typographical error. Additionally, "haematoxiline and eosine" on line 302 should be corrected to "haematoxilin and eosin," and "analyses" in Figure 1 should be amended to the singular form, "analysis."
All typographical errors have bene corrected and in particular those mentioned by the Reviewer.
- Тhere is a discrepancy in the frequency data. There appears to be an inconsistency regarding the reported frequency of DDR gene alterations in non-metastatic prostate cancer (nmPCa). While the text states a range of 5-10% (lines 103-104), the most frequently observed pathogenic variant, BRCA2, is cited as having a frequency of 5.3%, which barely exceeds the lower bound of this range.
The discrepancy has been corrected. BRCA2 frequency in non metastatic is 3.0%
3) Table 1.
- a) Column names may be given in full without a separate legend. Now names are given in full.
- b) The meaning of the 'familiarity' column is not clear; if it is information about a close relative with a similar cancer type or mutation in the DDR gene, this should be stated explicitly. Related information appears only 2 pages later on line 354 without mentioning the type of cancer and/or mutation.
The meaning of familiarity is that reported in the European Urological Association Guidelines 2024: “one or more first or second degree relatives with PCa on the same side of pedigree”. This is the only one parameter used to define familiarity on an anamnestic analysis.In table 1 and 2 cases are described for the presence or not of familiarity for PCa. No cases with familiarity for other cancers related to HRR mutations have been described. It is not Hereditary with the analysis of DDR gene mutation. This explanation is better reported in paragraph 4.1.1
- c) In the 'stage' column, 'metastatic' and 'non-metastatic' can be explicitly stated.
In the “stage” column, according to the TNM Staging System, the acronym M1 stands for metastatic cancer, while M0 for non-metastatic cancer
- d) No explanation of what 'A', 'B', 'C' slices mean.
There is no explanation. A,B,C is not a classification or an order for slides. Depending on the representative tissue available at biopsy, Pathologist prepared from one to three slices in each case named A,B,C without any classification or order. Cellularity is reported in each slide. Now this aspect is better explained in paragraph 4.1.2
- e) There is no explanation of what 'n.p.' means.
n.p. stands for “not performed”, as now fully reported in Table 2.
- f) How the variants of allele frequency were estimated?
According to the reviewer’s comment, we better explained how the VAF was estimated (paragraph 4.2. Findings). In brief, the Variant Allele Frequency (VAF) was estimated by the analysis software by counting, at the chromosomal position of the variant, how many reads for the mutated genotype are present over the total number of reads.
4) The ArcherDX analysis software has a reference to Figure 1 which does not explain what ArcherDX is.
The Archer Integrated DNA technologies is a company developing and producing reagents, platforms and softwares for molecular genetic analyses. The business location and the company references were added in the text paragraph 4.1.3
5) Figure 1 shows a 'Workflow for HRR analysis in metastatic prostate cancer'. Is this workflow only suitable for mPCa? In Tables 1,2, samples from nmPCa patients were analysed using a different workflow?
Figure 1 show in a very generic way without going intomethodological details the Workflow used for analysis. It is the same in both metastatic and non-metastatic cases and now this is clear from Figure 1 first line.
6) Also, comments on the proposed workflow:
- a) It seems to me that the quality of total genomic DNA isolation should be assessed by the presence of double-strand breaks by separating potential DNA fragments by gel electrophoresis. With qPCR, only the integrity of specific regions of the genome covered by primers can be assessed.
We agree with the reviewer that the gel electrophoresis is a potentially useful method for evaluating the DNA fragmentation. The method is qualitative rather than quantitative and has limited accuracy in DNA quantification, while the preparation of NGS libraries relies on precise quantification. The Archer PreSeq DNA QC Assay, a quantitative PCR (qPCR)-based method that was developed specifically for providing a quantitative assessment of DNA integrity, was adopted for this reason. This method enabled us to obtain a specific parameter, the DNA QC Score, which is linked to the amount of amplifiable genomic DNA. More in details, the Archer PreSeq DNA QC Assay employs a qPCR to identify DNA samples of sufficient quality to produce adequate sequencing library using Archer VARIANTPlex assays. It is a SYBR Green assay amplifying a 100bp genomic DNA sequence in the sample and in the Assay Standard in two separate reactions. A comparison of the two resultant quantification cycle (Cq) values results in a ΔCq, or DNA QC Score. The DNA QC Score holds predictive value for library yield and can be used to gauge the amount of input material required for successful Archer VARIANTPlex library preparation. The DNA QC Score obtained is a direct measurement of functional DNA templates, which provides an indicator of library yield and library quality metrics when using Archer VARIANTPlex assays. This is better described in paragraph 4.1.3.
- b) The 4th point is distributed between 3 and 5. NGS quality assessment is the quality control of fastq files. SNP variant analysis and classification is the final stage of SNP calling. If all the bioinformatic analysis is performed on a third party platform (such as ArcherDX) and not with your own tools, it is better to specify this explicitly.
Both the reads’ alignment over the reference genome and the variant calling were performed by a third party platform (i.e. ArcherDX analysis software). This information was added to the “4.1.3. Genetic Analysis” paragraph.
7) There is no way to trace associations between PV DDR gene, cancer type and other parameters from Table 2. It is probably redundant here.
We can agree with you but we believe that also the description of “ failed” cases in a separate Table ( Table 2) can be useful.
A few more comments on the paper as a whole:
The following considerations should be given when considering the overall manuscript:
- A comprehensive discussion of DDR genes is required. While the manuscript mentions various DNA Damage Repair (DDR) genes, and Table 1 specifically lists genes associated with particular patient pathology variants, the discussion in the second paragraph should be expanded to explore the roles of other DDR genes (at least those detailed in the table). This discussion should include an evaluation of their specific contributions to pathogenesis and suggest potential reasons for their association with particular types of pathology, thereby providing a more thorough overview of the DDR pathway involvement in the study.
Thank you for the constructive indication. As requested now Discussion paragraph 4.3 analyzes the different TIER I-III variants isolated in our population of non-metastatic PCa, reporting data in literature on their clinical significance. Limited data are present for PCa.
2) Justification of the Chosen Workflow: The presented workflow for HRR analysis does not appear to demonstrate a clear advantage over classical Sanger sequencing. It incorporates multiple comparable stages to those required for Sanger sequencing. Moreover, Sanger sequencing is a more cost-effective, accurate and time-efficient method for this application. Consequently, the manuscript should provide a more robust justification for the adoption of the chosen workflow, particularly in comparison to the established Sanger sequencing method, which may be more appropriate to the objectives of this study.
The aim of Figure 1 is to show in a very generic way, without methodologic details, the different steps used to prepare and to obtain analysis on HRR. The description does not go inside and does not describe the sequencing method used. Methods used are described in paragraph 4.1.2 and 4.1.3 pathologic and genetic analysis. In our opinion is a very simple workflow without specifying the use of different methods. We described the 4 steps that reader can find in the text in paragraph 4.1.2 and 4.1.3: First step: Sample preparation from Pathologist (described in paragraph 4.1.2) ; Second Step: DNA Extraction ( described in paragraph 4.1.3); Third Step: Next Generation Sequencing ( described in paragraph 4.1.3); Fourth Step: Analysis of variants ( described in paragraph 4.1.3). Is not the aim of this article to go inside a comparison of specific methods. We can also remove the figure, but in our pinion is simply descriptive.
Reviewer 4 Report
Comments and Suggestions for Authors
The article presents a comprehensive analysis of early detection of pathogenic variants of homologous recombination repair (HRR) genes in prostate cancer. The study examines the prognostic significance of these variants in newly diagnosed prostate cancer patients and their relationship with treatment strategies. However, the study may require more details, especially in the definition and methodology sections. Despite these shortcomings, the article makes a significant contribution to the field. It can be published after minor corrections.
I suggest some minor corrections.
1- More tables and graphs can be added to make the data easier to understand.
2- More detailed analysis of the differences in findings between patient groups can be done.
3- Detailing the findings for HRR gene variants other than BRCA2 can emphasize their impact on treatment strategies.
4- Technical details on DNA extraction from FFPE tissues and NGS library preparation can be clarified.
5- Selection of clinical samples and inclusion criteria can be defined in more detail.
Comments on the Quality of English LanguageSome paragraphs need to be made more fluent. Long sentences need to be shortened. Please have it checked by a native English speaker.
Author Response
Dear Reviewer, thank you for your positive and constructive comments and to give us the possibility to revise and improve our manuscript. We answered to your questions and now the manuscript improved. Modifications are evidenced in green.
I suggest some minor corrections.
- More tables and graphs can be added to make the data easier to understand.
We must respect the limits of length and number of tables and figures requested by the Journal and Editors for our paper. We have two long tables describing in details results and one figure describing methods.
- More detailed analysis of the differences in findings between patient groups can be done.
As requested differences in finding between patient groups ( in particular non-metastatic and metastatic PCa) are described not only in Table 1 but also in paragraph 4.2.
- Detailing the findings for HRR gene variants other than BRCA2 can emphasize their impact on treatment strategies.
We perfectly agree with you. At now in metastatic PCa, treatment strategy is decided mainly on the basis of BRCA analysis. The extention of analysis to a larger series of HRR variants could offer more information to select patients. This is not possible at now and our analysis is not able to give therapeutic implications. We now describe in ore details finding for HRR variants in our series ( paragraph 4.2) and we also discuss in more detain this aspect in paragraph 4.3 and conclusions
- Technical details on DNA extraction from FFPE tissues and NGS library preparation can be clarified.
The DNA extraction was performed by and an automated extractor and following the manufacture’s procedures, as described in the “4.1.3. Genetic Analysis” paragraph (evidenced either in green or in yellow). The process started with the preparation of Proteinase K, which was reconstituted in a buffer solution and stored at -20°C until needed. Tissue samples were placed in a cartridge well along with Sula Oil and Proteinase K, allowing the system's one-step heating method to simultaneously melt paraffin and lyse tissue. In the final step, genomic DNA is purified using cellulose-coated magnetic bead technology.The DNA libraries were prepared by an amplicon-based protocol by the use of the Archer™ VARIANTPlex™ Pan Solid Tumor Panel (Boulder, CO, USA) according to the manufacturer's instructions.
- Selection of clinical samples and inclusion criteria can be defined in more detail.
As requested, inclusion criteria and methods to select and analyze cases included is better described in paragraph 4 and 4.1.1
Round 2
Reviewer 3 Report
Comments and Suggestions for Authors
Dear Authors,
We would like to express our gratitude for the efforts you have made and for taking into account the previously provided comments. You have conducted significant work in revising the manuscript, and the current version appears greatly improved. Your responses to the earlier comments were detailed and precise, and the revisions have enhanced the clarity of the text, resolving most potential misunderstandings for readers.
Nevertheless, there remain a few aspects that require further refinement:
Lines 94–102:
The reported frequencies of mutations in DDR genes still require clarification. In their current form, the presentation of data may not be entirely clear to the reader. It is recommended, based on the data from reference 12, to specify more explicitly which sample proportions are presented in the table and how these relate to the reported frequency ranges.
Summary data visualization:
The "Discussion" section includes numerous statistical data regarding mutation frequencies and their associations with different forms of PCa. To improve the readability and visual representation of the information, we recommend adding a summary histogram showing the frequency of key pathogenic variants in patients with metastatic and non-metastatic PCa, and, if possible, relative to a control group.
Author Response
Dear Reviewer we sincerely thank you for your help and your constructive suggestions that strongly help us to significantly improve our manuscript.
We addressed your two remaining questions:
Lines 94–102:
The reported frequencies of mutations in DDR genes still require clarification. In their current form, the presentation of data may not be entirely clear to the reader. It is recommended, based on the data from reference 12, to specify more explicitly which sample proportions are presented in the table and how these relate to the reported frequency ranges.
We clarify the incidence of mutation in metastatic versus non metastatic PCa
Summary data visualization:
The "Discussion" section includes numerous statistical data regarding mutation frequencies and their associations with different forms of PCa. To improve the readability and visual representation of the information, we recommend adding a summary histogram showing the frequency of key pathogenic variants in patients with metastatic and non-metastatic PCa, and, if possible, relative to a control group.
We add a Table summarizing data on the DDR PVs discussed in this section.